# ‘Not to Be Harsh but Try Less to Relate to ‘the Teens’ and You’ll Relate to Them More’: Co-Designing Obesity Prevention Text Messages with Adolescents

**DOI:** 10.3390/ijerph16244887

**Published:** 2019-12-04

**Authors:** Stephanie R. Partridge, Rebecca Raeside, Zoe Latham, Anna C. Singleton, Karice Hyun, Alicia Grunseit, Katharine Steinbeck, Julie Redfern

**Affiliations:** 1Westmead Applied Research Centre, Faculty of Medicine and Health, The University of Sydney, Westmead, NSW 2145, Australia; rebecca.raeside@sydney.edu.au (R.R.); zoe.latham@sydney.edu.au (Z.L.); anna.singleton@sydney.edu.au (A.C.S.); karice.hyun@sydney.edu.au (K.H.); julie.redfern@sydney.edu.au (J.R.); 2Prevention Research Collaboration, Charles Perkins Centre, Sydney School of Public Health, Faculty of Medicine and Health, The University of Sydney, Camperdown, NSW 2006, Australia; 3Department of Weight Management, The Children’s Hospital Westmead, Westmead, NSW 2145, Australia; alicia.grunseit@health.nsw.gov.au; 4Discipline of Child and Adolescent Health, Faculty of Medicine, University of Sydney, Westmead, NSW 2145, Australia; kate.steinbeck@health.nsw.gov.au; 5The George Institute for Global Health, The University of New South Wales, Camperdown, NSW 2006, Australia

**Keywords:** adolescent, obesity prevention, text message, digital technology, mHealth, co-design, behavior change

## Abstract

Text messages remain a preferred way for adolescents to communicate, and recent evidence suggests adolescents would like access to digital healthcare options. However, there is limited evidence for text messages to engage adolescent populations in obesity prevention behaviors. We aimed to co-design a bank of text messages that are evidence-based, acceptable, and engaging for adolescents. An established iterative mixed methods process, consisting of three phases, was used to develop the text message program. The first bank of 145 text messages was drafted based on current evidence, behavior change techniques, and input from researchers and health professionals. A survey was then administered to adolescents and professionals for review of text message content, usefulness, understanding, and age-appropriateness. An adolescent research assistant collaborated with the research team on all three phases. Forty participants (25 adolescents and 15 professionals) reviewed the initial bank of 145 text messages. On average, all reviewers agreed the text messages were easy to understand (13.6/15) and useful (13.1/15). In total, 107 text messages were included in the final text message bank to support behavior change and prevent obesity. This study may guide other researchers or health professionals who are seeking to engage adolescents in the co-design of health promotion or intervention content. Effectiveness of the text message program will be tested in a randomized controlled trial.

## 1. Introduction

In 2016, an estimated 337 million children and adolescents 5 to 19 years of age had overweight or obesity [1]. Concerningly, it is estimated that 70% of children and adolescents with obesity between the ages of 5 and 17 years have at least one risk factor for cardiovascular disease [2]. Effective prevention of obesity is essential during adolescence because at least 90% of adolescents with obesity will have overweight or obesity in young adulthood [3,4]. Adolescents with overweight or obesity are also at a higher risk of adverse health consequences throughout adulthood [5,6,7].

A recommended strategy to prevent obesity in adolescents is lifestyle changes, including reduced energy intake and sedentary behavior, increased physical activity, and measures to support behavioral change [8]. To date, most interventions have been delivered in-person [9], which may decrease access for many adolescents at risk of obesity [10]. In obesity prevention and management intervention studies, there is limited evidence of adolescent involvement, such as in the co-design of intervention content [11,12]. Current evidence also suggests that when adolescents are engaged in intervention design it is often ad hoc and tokenistic [13]. Adolescents are immersed in a digital world and the ubiquity of digital technology, such as mobile phones, offers a potential opportunity for acceptable and accessible obesity prevention interventions [14]. Given this, digital interventions that are co-designed with adolescents may increase intervention engagement and efficacy, and research is required to expand the evidence base to provide accessible prevention options.

Amongst all smartphone capabilities, text messages remain a preferred form of communication for adolescents to communicate with their peers [15]. Text messages do not require an internet connection nor do they incur a cost to receive, thereby increasing accessibility and affordability and offering a socially equitable and novel way to deliver health promotion interventions [16]. Personalized health promotion interventions sent by text message are recognized as a useful tool for chronic disease risk factor prevention and management [17,18,19,20]. Yet, there remains limited evidence of text messages to support adolescents at risk of obesity [21,22]. Our recent review found only eight interventions that utilized text messages for obesity prevention and management in adolescents [23,24,25,26,27,28,29,30]. Only three of the eight studies engaged adolescents in the text message development process [23,25,26]. However, none of the three studies provide explicit detail about the co-design process with adolescents regarding the text message content and language.

The text message program behavioral intervention for teens on eating, physical activity, and social wellbeing (TEXTBITES) is a mobile health (mhealth) promotion intervention co-designed with adolescents (13–18 years of age) [31]. The intervention group receives a six-month text message program, which consists of interactive, semi-personalized, lifestyle-focused text messages (four text messages per week) with optional telephone health counselling (one telephone call per month). The study also includes a follow-up at 12 months. Intervention efficacy is currently being tested in a randomized controlled trial (RCT) design. The primary outcome of the program is change in body mass index z-score at six months. The TEXTBITES study is one of the first few co-designed text message programs with adolescents. We conducted iterative mixed methods research to develop the text message component of this study. Specifically, this study aimed to co-design the bank of text messages that are evidence-based, acceptable, and engaging for adolescents, 13 to 18 years of age.

## 2. Materials and Methods

### 2.1. Design

A co-designed bank of text messages was developed using a previously published iterative scientific process using a mixed methods study design [32] involving adolescents, health professionals, and researchers [33]. Data were collected in three phases: phase 1, initial development phase; phase 2, user acceptance testing; and phase 3, final development and platform testing. The study was conducted between December 2017 and August 2019. Ethics approval was obtained from the Sydney Children’s Hospital Network Human Research Ethics Committee (Approval Number: HREC/18/SCHN/374).

### 2.2. Phase 1: Initial Development of Text Messages

The aim of phase 1 was to develop the first bank of text messages for user acceptance testing in phase 2. This phase involved several strategies, including (i) identifying key behaviors associated with obesity in adolescents; (ii) a narrative review of previous text message development work with adolescents and behavioral change techniques (BCTs); (iii) a 2-hour workshop with researchers and health professionals with experience working with adolescents to form a consensus on key text message content areas, frequency and timing of text messages, and drafting initial text messages; and (iv) a review of initial text messages and drafting of new text messages by a 19-year-old adolescent research assistant employed by the research team. The adolescent research assistant completed research training specific to the current study offered by the research team and general research training offered by the university.

### 2.3. Phase 2: User Acceptance Testing

The aim of phase 2 was to test the content of the draft text messages using a qualitative user survey based on a previous research study [32] and to assess readability level of the text messages. Each survey included two questions that required Likert responses about understanding and usefulness with responses assigned a score out of five (strongly agree = 5; agree = 4; neutral = 3; disagree = 2; or strongly disagree = 1). Each survey also included one question about age-appropriateness, where respondents could tick none or all ages that applied (response options included: 13–14 years of age; 15–16 years of age; or 17–18 years of age). A final open-ended question asked for general comment, feedback, and any suggestions for improvement. Demographic characteristics were also collected from adolescents and included age, gender, postcode (for categorizing socio-economic status), language spoken at home, education level, and current school year. Demographic characteristics collected from professionals included age, gender, and area of expertise.

Each text message was reviewed six times by three adolescents and by three professionals with experience working with adolescents. Each participant reviewed between 15 to 30 individual text messages. Adolescent reviewers were included if they were (i) 13–18 years of age and (ii) provided written informed consent and, if <18 years of age, additional written informed consent from their parents or guardians. Adolescent reviewers were excluded if they had (i) a medical condition or psychiatric illness that would not allow the participant to give informed consent and/or would preclude the participant’s ability to comply with the study protocol; (ii) a history of disordered eating, including being diagnosed with, or treated for, anorexia nervosa or anorexia athletica, binge eating disorder or bulimia nervosa; or (iii) inability of the participant to speak English. Adolescents were recruited from an outpatient weight management clinic in a public hospital or from the wider community via emails to youth groups in the local health district and youth group contacts from the research team’s networks. Professional reviewers included (i) multidisciplinary health and research professionals and (ii) who provided written informed consent. Professional reviewers were recruited via email invitation from the research team’s networks. All surveys responses were de-identified.

Total scores (out of 15 points) for understanding and usefulness were calculated separately for adolescents and professionals by summing the scores of each reviewer (out of 5 points). Text messages that scored greater than or equal to 12 points by adolescents and professionals for both understanding and usefulness were included. Text messages which scored a total of less than 12 points by adolescents and professionals for understanding or usefulness were excluded. All included open-ended feedback was collated and summarized. Suggestions or concerns were addressed, and the included text messages were modified accordingly by the adolescent research assistant in collaboration with other members of the research team. The updated bank of text messages was then ready to be finalized and tested using the commercial delivery platform.

### 2.4. Phase 3: Final Development and Platform Testing

The aim of phase 3 was to consolidate the findings of phase 1 and 2, develop a final bank of text messages for use in a six-month behavior change intervention, and test their delivery using a text message platform for business marketing repurposed for use in healthcare. This phase involved determining the readability level of the text messages and the text message schedule, which included ordering and timing of the text messages. The Flesch–Kincaid readability ease score was used to assess the readability level of each text message [34]. The score represents the approximate education level a person will need to be able to read a particular text easily. The score considers the number of syllables per word and the number of words per sentence. Each text message was inserted into a predetermined equation to determine readability ease score from 0 to 100; 0 being the most difficult to read and 100 being the easiest to read. All text messages were scored. The scheduling process was finalized in collaboration with the adolescent research assistant. The final text message schedule was reviewed by the research team (authors). Finally, platform testing was performed. Text messages were programmed into the commercial platform (Burst SMS) and pre-scheduled for delivery over seven days. Six mobile phone users from the research team with different operating systems (e.g., iOS, Android) elected to receive messages. Users were provided with a log sheet to record the date and time messages were delivered and any issues. Users were also asked to reply to one to two text messages over the seven days to determine how replies would appear on the online user dashboard.

## 3. Results

### 3.1. Phase 1: Initial Development of Text Messages

#### 3.1.1. Key Behaviors Associated with Obesity in Adolescents

Several behavioral risk factors are related to excess weight gain and obesity in adolescents. Data from the National Health Survey in 2017–2018 demonstrate most adolescents are not meeting national recommendations for physical activity and sedentary behaviors or fruit and vegetable consumption [35]. Only 10.5% and 22% of 13 to 14-year-olds are meeting physical activity and sedentary behavior guidelines, respectively, and compliance decreases with age, with 5.5% and 19% of 15 to 17-year-olds meeting physical activity and sedentary behavior guidelines, respectively [35]. Less than 4% of adolescents eat enough fruits and vegetables [35]. Additionally, adolescents are the highest consumers of discretionary foods and sugar-sweetened beverages [36].

Sub-optimal lifestyle risk factors during adolescence are associated with increased risk of non-communicable diseases. Evidence from 20 unique prospective observational studies suggests that increased physical activity and decreased sedentary behaviors are protective against relative weight gain and adiposity over adolescence [37]. The evidence for consuming adequate fruit and vegetables and weight regulation, in isolation from lower energy intake or increased physical activity, remains unclear in adolescents [38,39]. However, diets high in fruits and vegetables are essential for overall health and are independently associated with a decreased risk of other non-communicable diseases, such as cardiovascular disease, stroke, and some cancers [40]. Increased physical activity levels and decreased sedentary behaviors and consuming adequate fruit and vegetables are associated with the prevention of excess weight gain and adiposity in adults [41,42].

Excess consumption of sugar-sweetened beverages (SSB) and discretionary foods (foods that are not required for a healthy diet, characterized by a high composition of saturated fat, added sugar, or added salt) have been associated with increased obesity risk in adolescents [43,44]. Data from the Coronary Artery Risk Development in Young Adults (CARDIA) study, which followed 3031 young adults over 15 years of age, found high consumptions of SSB and discretionary foods have strong, positive, and independent associations with weight gain and insulin resistance [45]. SSB and discretionary foods consumption behaviors are often well-established during adolescence. Accordingly, a significant proportion of adolescents enter young adulthood at a much higher risk of obesity than when they entered adolescence [46]. Therefore, adolescence is a critical period for intervention. A randomized controlled trial (RCT) designed to decrease consumption of SSB in 224 adolescents with overweight and obesity demonstrated significantly lower increases in BMI in the intervention group compared to the control group at 12 months [47]. Adolescents in the intervention group, who were provided with noncaloric beverages every 2 weeks, significantly decreased their SSB consumption to nearly zero compared to the control group [47]. Emerging research suggests that, to improve an individual’s diet quality, intervention components should focus on synergistic strategies to increase core food consumption and reduce the intake of discretionary foods, particularly SSB consumption [48].

Other behavioral risk factors are associated with obesity in adolescents. Insufficient sleep duration and poorer sleep quality are associated with a higher BMI and cardiometabolic risk, including insulin resistance, dyslipidemia, and higher blood pressure in adolescents [49]. Stressful major life events and negative body image (independent of depression) may be associated with obesity in adolescents [50,51]. Recent research has also suggested being labelled as ‘too fat’ at 14 years of age increases the risk of obesity in late adolescence and early adulthood [52]. It has been proposed that a weight stigma influences health by increasing stress, unhealthy behavior changes, disengagement with health care services, and social withdrawal [53].

#### 3.1.2. Previous Text Message Development Work with Adolescents

Five previous studies have reported results of text message development or process research with adolescents for obesity prevention behaviors [54,55,56,57,58] (Table 1). All three development studies found adolescents were enthusiastic and excited about receiving text messages about nutrition and physical activity behaviors, food literacy, and weight management [54,57,58]. Adolescents preferred text messages with an active voice, a reference to adolescents, and recommendations for specific and achievable behavior change. The findings reinforced that adolescents are sensitive to language and that text messages should avoid an authoritarian tone, which may induce feelings of shame. The findings from the studies highlight the importance of addressing barriers to behavior change, such as time management and navigating food and activity environments. Moreover, to increase engagement, text messages can potentially address issues of concern to adolescents related to their health behaviors, such as climate change and plastic use.

#### 3.1.3. Effective Behavioral Change Techniques for Obesity Prevention in Adolescents

A systematic review of 17 childhood obesity management and prevention behavior change interventions has previously been conducted, which identified effective BCTs [59]. The authors of this systematic review used the behavior-specific taxonomy of 40 BCTs for physical activity and healthy eating behaviors (CALO-RE taxonomy) [60]. The systematic review identified six BCTs that may be effective components of obesity management interventions, namely, provide information on consequences of the behavior to the individual; environmental restructuring; prompt practice; prompt identification as a role model/position advocate; stress management/emotional control training; and general communication skills training [59]. One effective BCT for obesity prevention interventions was identified, namely, prompting generalization of a target behavior [59]. The first bank of text messages was developed focusing on these seven BCTs, referring to the definitions [60].

#### 3.1.4. Workshop with Health and Research Professionals

Eleven multidisciplinary health and research professionals, with expertise in obesity, cardiovascular disease, nutrition, physical activity, mental health, and public health, participated in the 2-h workshop. The researchers agreed on four content priority areas based on the latest evidence of key behaviors associated with obesity in adolescence, namely, physical activity, nutrition, mental wellbeing, and general behaviors (Figure 1). It was agreed that the text messages would be interactive, offering two-way communication with a health professional, and four text messages would be sent each week, including one weekend day at random times, using previous evidence from text message development work with adolescents. Text messages were scheduled for delivery at random times to reduce pattern detection and increase engagement, and the program would have an active intervention period of six months.

#### 3.1.5. Initial Text Messages Bank Drafting

During the workshop, the multidisciplinary researchers drafted 300 text messages. After the meeting, all text messages were reviewed by the lead author and adolescent research assistant, similar messages deleted, and new messages drafted in collaboration with the adolescent research assistant. The initial bank consisted of 145 text messages. All text messages were based on the four content priority areas, available evidence-based health information found in current national guidelines, effective behavior change techniques, and previous text message development findings.

### 3.2. Phase 2: User Acceptance Testing

#### 3.2.1. Participant Demographics

In total, 40 participants reviewed the initial bank of 145 text messages (Table 2). The sample included 15 professionals, who were predominantly female (12/15), mean age of 36 years (standard deviation (SD) 11), with varying areas of expertise, and 25 adolescents, who were predominantly female (15/25), mean age of 15 years (SD 2), predominately English-speaking at home (21/25), and living in a socioeconomically advantaged area (quintile 5, 11/25).

#### 3.2.2. Text Message Understanding and Usefulness Scores and Feedback

On average, all reviewers of the 145 text messages agreed that the text messages were easy to understand (13.6/15) and useful (13.1/15). In total, 105 text messages were included in the final text message bank (rating on average 14/15 for understanding and 13.6/15 for usefulness). No differences in scores were observed between the different text message content areas. Forty text messages were deleted (12 nutrition messages, 8 physical activity messages, 8 general behavior messages, and 12 mental wellbeing messages). The reasons for inclusion, exclusion, or modification of text messages are summarized in Table 3. Text messages that rated highly were practical and fun, focused on the environmental impacts of eating and activity, highlighted benefits other than physical health, and were succinct and straightforward. Common modifications included ensuring the text messages were age appropriate. Fourteen of the 105 text messages were rated appropriate for only older adolescents, 17–18 years of age and 8t text messages were rated suitable for only younger adolescents, 13–16 years of age. New age-appropriate text messages were created for each of these 22 text messages.

Further modifications included removing youth jargon, abbreviations, or unnecessary emojis, limiting puns and jokes, and providing more practical advice and links. Besides, editorial changes were made to improve sentence structure, punctuation, and style. Forty text messages were rated <12 by adolescents and professionals for understanding or usefulness and were excluded. The three most common reasons for exclusion were the text message provided advice that was different from the ‘norm’ of being a young person, text messages with conflicting scores between professionals and adolescents, messages that were perceived as ‘trying too hard’, and messages that were long or wordy. Introductory and final text messages were added.

### 3.3. Phase 3: Final Development and Platform Testing

#### 3.3.1. Final Text Message Bank

The final bank of 107 text messages included an introductory and final text message, 26 nutrition messages, 18 physical activity messages, 34 general behavior messages, and 21 mental wellbeing messages. Most text messages addressed two BCTs, with 45 text messages prompting the adolescent to repeat the target behaviors (prompt practice) and 21 text messages providing tailored information about the benefits and costs of action to adolescents (provide information on consequences of the behavior to the individual). Text messages addressed environmental restructuring (n = 14), stress management or emotional control training (n = 14), instructions on how to perform the behavior (n = 14), time management (n = 9), and prompting identification as a role model (n = 8). Twenty-two text messages encouraged two-way communication via quizzes or short questions. Six specific text messages encouraged interaction with the health counsellor (sent once per month). Twenty-two text messages differed based on age (13–16 or 17–18 years of age). As such, two separate age-appropriate sequences of the 107 text messages were created. The mean Flesch–Kincaid readability ease score of the 107 text messages was 76.1 (SD 12.1), indicating the text message bank, on average, is at a seventh-grade reading level (13 years of age) and classified as ‘fairly easy to read’. Most text messages (n = 74) were rated as ‘very easy to read’ to ‘fairly easy to read’ (fifth grade: 11 years of age, sixth grade: 12 years of age, or seventh grade: 13years of age reading level). Thirty-three (19 and 14 text messages, respectively) text messages were at an eighth to ninth grade (13 to 15 years of age) or tenth to twelfth grade (16 to 18 years of age) reading level.

#### 3.3.2. Platform Testing

Platform testing occurred in conjunction with another text message study. Thirty-two text messages from both programs were pre-scheduled into the platform. All text messages were delivered successfully and no issues were identified. Replies were received in the form of text and emojis. All types of replies were easily viewed on the online user dashboard. Data can be downloaded from the user dashboard as a comma-separated value (CSV) file for analysis of text messages sent and received.

## 4. Discussion

In this study, an iterative, co-design approach was used to engage adolescents in developing and refining text messages, resulting in a bank of 107 text messages that are evidence-based, acceptable, and engaging for adolescents. In addition, the text messages are at an appropriate literacy level for adolescents to ensure they are understood. This study worked with adolescents from a weight management clinic, the wider community, and an adolescent research assistant, who was a member of the research team and who was engaged in all three phases of research. This process endeavored to ensure adolescent views and interests were at the forefront of the text message development process. This approach was feasible and resulted in useful feedback that substantially improved the initial bank of text messages. Furthermore, the text messages were developed based on behavior change techniques with demonstrated effectiveness in the prevention and management of adolescent obesity [59]. Overall, user acceptance testing found adolescents preferred text messages that were practical and fun, focused on environmental sustainability, highlighted benefits other than physical health, and text messages that were succinct and straightforward.

A review of eight studies investigating weight management interventions for adolescents using text messages for program delivery, found that studies report outcomes rather than the process of text message development [23,24,25,26,27,28,29,30]. There are limited studies investigating weight management programs delivered exclusively by text message. Most studies are multicomponent, with limited process evaluation data to elucidate adolescents’ perceptions of and engagement with text messages. Key findings from the five published studies to date, which report on either text message development or process research with adolescents for weight management or behavior change, were used in developing the initial bank of text messages for the current study [54,55,56,57,58]. The findings from these 5 studies, particularly related to language, tone, positivity, and practicality of text messages, were incorporated into the initial text message bank. The overall high scores for understanding (13.6/15) and usefulness (13.1/15) received in the user acceptance phase of the current study may be explained by this comprehensive review of prior work and application of these findings to the initial text message bank.

In addition to the review of prior text message development work and application of the findings, we included an adolescent research assistant as a member of the research team. Research and guidelines for the prevention and management of obesity in adolescents are still almost entirely driven and implemented by adults [8]. A 2019 systematic review of 65 frameworks for consumer involvement in research found limited evidence of generalizability and that a range of evidence-based consumer frameworks is required for different stakeholders in research [61]. Though, of the 65 frameworks reviewed, none were specific to the unique needs of adolescents. There is limited guidance available to researchers about how to effectively engage adolescents in research on health issues that affect them, particularly for the prevention of risk factors for chronic disease, such as obesity. A recent study with young adults, 18–24 years of age used an online survey to identify social media messaging types preferred by young adults to improve their calcium intake using pre-developed content. The study found young adult participants recommended hiring young people to develop the social media content [62]. The employment of an adolescent research assistant for the current study provided unique views and insights about how to best communicate with adolescents.

In the present study, the adolescent research assistant led the development of text messages related to social issues of concern to adolescents, including climate change and plastic use. Text messages addressing these current social issues were rated highly by adolescents in the user acceptance testing phase. Such social issues shared similar behavioral goals with those for obesity prevention and may motivate adolescents to change dietary and physical activity behaviors to help prevent climate change and reduce plastic pollution, not for purposes of improving their health behaviors or preventing obesity [63]. There is evidence to suggest that, from a young age, adolescents have high levels of environmental awareness and basic comprehension of complex environmental issues [64]. Therefore, it is hypothesized sustainability messages may help educate and motivate adolescents to make healthier physical activity and food choices.

Although overall the text messages appealed to adolescents, it remains unclear whether the text messages, once received as part of an overall program, will result in changes in nutrition and physical activity behaviors and lead to weight changes. BCTs support the text messages with demonstrated effectiveness in adolescent obesity prevention and management interventions [59]. To date, there has been limited research to identify the mechanisms underpinning obesity-related behavior change in the adolescent population. The seven BCTs selected for the text messages were based on one systematic review, which included the intervention content of 17 individual intervention studies assessing BMI outcomes at six months [59]. The authors of this systematic review applied a stringent coding strategy, however, it is possible that additional effective BCTs were not coded due to insufficient descriptions in the study publications. In the current ongoing study, additional support for behavior change will be provided through communication with the health counsellor using complementary theoretical approaches with evidence for behavior change, including motivational interviewing, goal setting, self-monitoring, and barrier identification and problem solving [65,66]. The efficacy of the text message program is currently being tested in an RCT [31]. A comprehensive process evaluation will also be conducted to understand how adolescents engage with the text messages and the health counsellor and how behavior change at the individual level is impacted by the adolescents’ broader social, economic, and political environments. However, individual behavior change programs are necessary components of larger systems to prevent obesity [67].

This study demonstrated a novel way to engage adolescents in research on health issues that affect them. However, there is limited guidance and research about the most effective strategies to engage adolescents in the co-design of research. Further research is required to develop effective co-design strategies, particularly for the prevention of risk factors for chronic diseases, such as obesity. Each individual text message was reviewed by six participants. This comprehensive approach was adapted from an effective text message study for adults with heart disease [32], which demonstrated a significant reduction in their BMI [17]. However, in the current study the sample of adolescents was recruited from an adolescent weight management clinic and the wider community. In the survey, no question regarding recruitment method was included and no anthropometric data were collected. Therefore, it cannot be determined if the views of adolescents with overweight or obesity were different from adolescents without overweight or obesity. More adolescents and professionals who participated in the review process identified as girls or women, respectively. This difference may have introduced sub-conscious gender biases to the text message content and may impact the program effects for participants identifying as boys. Due to the small sample size, no sub-group analyses were conducted to determine if responses to text messages differed by gender. Also, the sample of adolescents was recruited via convenience sampling. Recruitment was conducted in a diverse region of Sydney, Australia. However, there is a risk the text message reviews of adolescent participants may not represent the views of the wider adolescent population. Recruitment of adolescents with overweight for the RCT is being conducted from the wider community and the comprehensive process evaluation will provide wider views on the acceptability of the text message program in this population at greater risk of obesity.

## 5. Conclusions

In conclusion, this study reports on the development of a bank of 107 text messages that are now suitable for efficacy testing in an RCT. The text message bank was developed using current evidence, behavior change techniques, and a comprehensive adolescent and professional review phase. This study may guide other researchers or health professionals who are seeking to engage adolescents in the co-design of health promotion content. Text messaging interventions for adolescent obesity prevention have the potential to provide accessible and affordable health promotion services to all adolescents, regardless of geographical location or socioeconomic status.

## Figures and Tables

**Figure 1 ijerph-16-04887-f001:**
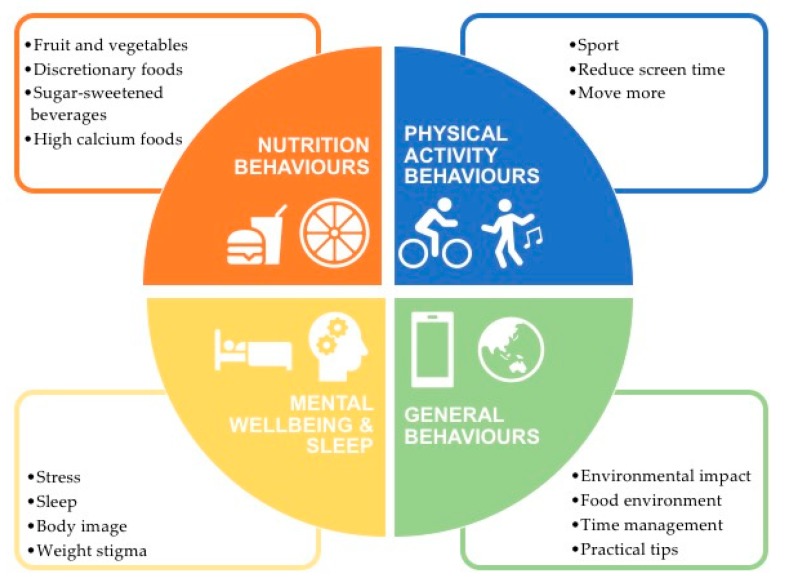
Content priority areas for text message for adolescents with overweight.

**Table 1 ijerph-16-04887-t001:** Previous text message development or process research work with adolescents.

Author, Year, Country, Citation	Study Type, Focus, & Design	Participants	Key Findings Related to Text Message Preferences
Hingle et al. 2013, US, [54]	Type: Formative research & pilot study Focus: Nutrition and physical activity behaviors Design: Youth participatory approach with focus groups, discussions, & 8-week pilot study	N = 177 (aged 12–18 years) focus groups, n = 59 discussions, n = 86 pilot study, n = 32	Active voice and no authoritarian tone Reference to adolescents Recommendations for specific and achievable behavior change from a nutrition professional Factoids (information in <160 characters) or quizzes ≤2 text messages/day
Smith et al. 2014, Australia, [55]	Type: Qualitative study Focus: Nutrition and physical activity behaviors for weight management Design: Focus groups with adolescents who had participated in an 8-week intervention and maintenance phase with text message support	N = 12 (aged 12–16 years)	Preferred more casual and personalized text messages Less frequent (3/week too frequent) Include reasons for wanting to change their behaviors Include practical tips Need to address barriers to behavior change, including time and tiredness Not motivated by testimonials from other adolescents
Thompson et al. 2016, US, [56]	Type: Feasibility study Focus: Physical activity behaviors Design: Four-group (40 per group) randomized design & post-intervention research with a subset of participants examined program reactions	N = 160 (aged 14–17 years) 20 in group receiving text messages completed post interviews and surveys	80% liked receiving text messages daily 75% rated them as helpful 55% said they were motivational Suggested adding variety and making less repetitive Text messages were sent at 08:00 and 55% of adolescents suggested sending at 15:00–18:00
Wickham and Carbone, 2018, US, [57]	Type: Formative research Focus: Food literacy Design: Community based participatory research approach with a Kid Council	N = 4 (aged 13–16 years) Formed Kid Council	Direct, fun, and straightforward Inclusive of fun facts and emojis Disliked abbreviations
Woolford et al. 2011, US, [58]	Type: Formative research Focus: Weight loss Design: Focus groups	N = 24 (aged 11–19 years)	Direct and told them what to do, particularly recipes Testimonials for successful weight loss strategies from their peers Positive and encouraging Natural tone and avoid colloquial abbreviations Contained emojis to convey enthusiasm Disliked when asked them to reflect on ways to make healthier choices Disliked content that may trigger unhealthy behaviors, such as mentioning unhealthy foods or sedentary behaviors

**Table 2 ijerph-16-04887-t002:** Participant characteristics (n = 40).

Characteristic		Participants
*Adolescents*		*n = 25*
Age (years ± SD)		15 ± 2
Gender identity	Female	15
	Male	10
Language spoken at home	English-speaking ^a^	21
Current high school student	Yes	25
Current school year	Years 7–8	6
	Years 9–10	13
	Years 11–12	10
SES quintile	0–60 (lowest) ^b^	8
	61–80	5
	81–100 (highest)	11
*Professionals*		*n = 15*
Age (years ± SD)		36 ± 11
Gender identity	Female	12
	Male	3
Area of expertise	Physical activity	1
	Nutrition and diet	4
	Adolescent medicine	2
	Medicine	2
	Public health	6
	Prevention	2
	Behavior change	4
	Psychology	1
	High school teacher	2

SES, socioeconomic status; SD, standard deviation. ^a^ Other languages spoken at home included Tamil, Arabic, Burmese, and Bahasa Indonesia. ^b^ Lowest three SES quintiles combined.

**Table 3 ijerph-16-04887-t003:** Sample reasons for text message inclusion, exclusion, or modification, categorized by main reasons for rating.

Original Text Message	Reasons for Rating	Rational for Rating	Amended or New Message
Corn isn’t just a tasty snack, it’s multi-purpose! It can be used to make fireworks, glue, paint & plastic. But let’s face it, popcorn is one of the best uses, check out some recipe ideas here: tinyurl.com/airpopcorn	Practical and fun	Scored 15/15 for both usefulness and understanding by both adolescents and professionals Adolescent reviewer: “Perfect.” Professional review: “This is a fun one!”	No revision
Producing a fast-food burger can create up to 3.5 kg of carbon emissions! Skip the processing, help save the planet & make your own: tinyurl.com/tanburger	Focused on environmental sustainability	Scored >12/15 for both usefulness and understanding by both adolescents and professionals Professional reviewer: “I really like this one.”	No revision
Need a dose of some happy hormones? Stretching can release endorphins, reduce your stress, make you feel great, and you can do it anywhere, even while watching TV or YouTube. Check it out: tinyurl.com/stretchyout	Benefits other than physical health	Scored 15/15 for both usefulness and understanding by both adolescents and professionals Professional reviewer: “I like this message! The second half could flow a bit better.” Revision: Modification to sentence structure	Need a dose of some happy hormones? Stretching can release endorphins, reduce your stress and make you feel great. The best part? You can do it anywhere, even while watching TV or YouTube. Check it out: tinyurl.com/stretchyout
Want to get your homework or study done in record time? Do power bursts! Put your phone on do not disturb and power it out for 25 min. Take a 5 min break (yep, you can check your phone) and repeat until it’s done!	Succinct and straightforward	Scored 15/15 for both usefulness and understanding by both adolescents and professionals Adolescent reviewer: “Good info!!” Professional reviewer: "I like this one, very straightforward. Maybe right at the end say, ’your work’ instead of ’it’s’."	Want to get your homework or study done in record time? Do power bursts! Put your phone on do not disturb and power it out for 25 min. Take a 5 min break (yep, you can check your phone) and repeat until your work is done!
Learning to cook now is the best time because, by the time you move out of home, you’ll be a whiz! It will save you money, keep you energized, as well as impress your friends & maybe someone special-a triple win! vegpower.org.uk/recipes/	Age-appropriate	Scored >12/15 for both usefulness and understanding by both adolescents and professionals This message was rated more appropriate for older adolescents, 17–18 years of age and an alternate text message was included for younger adolescents, 13–16 years of age. Revision: Included an age-appropriate message for younger adolescents.	Learning to cook now is the best time because it can help you understand different cultures and flavors! Check out some cool recipes here-vegpower.org.uk/recipes/
33% of all the world’s food gets wasted! Isn’t that crazy?! :O #FightFoodWaste & check out what you can do: tinyurl.com/fightfoodwa	Remove youth jargon, abbreviations, or unnecessary emojis	Scored >12/15 for both usefulness and understanding by both adolescents and professionals Professional reviewer: “Using hashtags in a text message won’t lead anywhere as they would in social media and could result in just being confusing. Instead, maybe direct the person to use the hashtag to display their actions to affect food waste.” Adolescent reviewer: “No emoticons, do you need the hashtag?" and “Yeah, I [don’t know] what that emoji means. Not to be harsh but try less to relate to ’the teens’, and you’ll relate to them more” Revision: Removed hashtag and emoji.	33% of all the world’s food gets wasted! Isn’t that crazy?! Fight food waste & check out what you can do to help: tinyurl.com/fightfoodwa
An egg, the most-liked photo on Instagram! No yolks, it was eggcellent. For a soft-boiled egg, cook for 5 min & hard-boiled eggs, cook for 8 min. Delicious on a wholemeal cracker or toast. For more tips check out this clip: tinyurl.com/eggboil	Limiting puns and jokes	Scored >12/15 for both usefulness and understanding by both adolescents and professionals Adolescent reviewer: “Too many puns?” Professional reviewer: “I’d stick to only using one pun in the message instead of 2.” Revision: removed pun ‘eggcellent.’	An egg is the most-liked photo on Instagram! No yolks! For a soft-boiled egg, cook for 5 min & hard-boiled eggs, cook for 8 min. Delicious on a wholemeal cracker or toast. For more tips check out this clip: tinyurl.com/eggboil
Ever pulled an all-nighter to study for an exam? If yes, don’t worry, you’re not alone! But lack of sleep will decrease your performance the next day. You’ll study better & retain more knowledge when you study consistently in small bite-sized chunks. Plan a study schedule for your next big exam!	More practical advice and links	Scored >12/15 for both usefulness and understanding by both adolescents and professionals Adolescent reviewer: "Maybe put a link to a weekly planner page to help with organisation" Revision: included link to an online study schedule	Ever pulled an all-nighter to study for an exam? If yes, don’t worry, you’re not alone! You’ll study better & retain more knowledge when you sleep well & study consistently in small bite-sized chunks. Plan a study schedule for your next big exam, try this template: tinyurl.com/y4ymrej2
Instead of binging on your fav TV show to avoid a finishing a big assessment, reward yourself with one episode after completing small chunks	Advice that is different from ‘norm’ of being an adolescent	Scored >12/15 for understanding by both adolescents and professionals and conflicting reviewer scores and feedback with adolescents rating 11/15 and professionals 15/15 for usefulness. Adolescent reviewer: “The nature of tv shows is that you feel compelled to binge them. watching one can end up being a whole series." Professional reviewer: “Excellent message, really important.”	Excluded
On a ‘kite’ study schedule & can’t find time for physical activity? Why not fly a kite! Kite flying is a professional sport in Thailand & it requires mental skills too like concentration	Perceived as ‘trying too hard.’	Scored <12/15 for both usefulness and understanding by both adolescents and professionals Adolescent reviewers: “I don’t know what a kite study schedule is, and I don’t believe that kite flying is teenagers preferred pastime” and “Not relevant, a bit weird.”	Excluded
Our brains weigh about 1.5 kg & are 70% water! The other 30% is made up of fat & protein with a little sugar & a little salt. Similar to pancakes, but more useful for thinking & far less delicious than pancakes. Speaking of pancakes: tinyurl.com/pancaketip	Long and wordy	Scored <12/15 for both usefulness and understanding by both adolescents and professionals Adolescent reviewer: “Slightly unclear.”	Excluded

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
