# Peer review of "‘Not to Be Harsh but Try Less to Relate to ‘the Teens’ and You’ll Relate to Them More’: Co-Designing Obesity Prevention Text Messages with Adolescents"

_ijerph, 2019, doi:10.3390/ijerph16244887_

Round 1
Reviewer 1 Report
Overall a nice study and generally well presented manuscript that details a novel approach to developing a text messaging-based system to promote healthy behaviors, especially proper nutrition. Good job.
The primary concerns with this manuscript is that it needs another round of thorough editing, and perhaps consideration of adding a bit more analysis related to the distribution of the 107 messages across the 4 content priority areas.
1. There are numerous typographical and grammatical errors throughout, including in Tables 1 and 3. These need to be cleaned up.
2. Table 2. Participant characteristics (n=40)
The line that reads: "Ethnicity English speaking(a) 21"
It seems odd that Ethnicity and English language proficiency are treated as the same. Perhaps the label Ethnicity should be changed. Also, in the text, English language proficiency was identified as an inclusion criterion, but in the table only 21 of the 25 adolescents are identified as speaking English, with a footnote identifying other languages spoken at home. There is a mismatch between each item on this line of the table, and a mismatch with the main text. Needs to be fixed.
3. Figure 1 does a nice job of representing the 4 main content priority areas for the text messaging intervention. It would be additionally helpful to see a table or other representation of how the 107 messages in the final selection are distributed across these 4 areas, and how the rejected messages are distributed. Might shed some light on the relative difficulty of messaging in each of these 4 areas.
4. The citation at the very beginning seems odd. It does not say where these 42 million children are. There are over 13.7 million overweight or obese children and adolescents under 19 years of age just in the US. The WHO estimates that there were 340 million children and adolescents worldwide in 2016 who were overweight.
https://www.cdc.gov/obesity/data/childhood.html
https://www.who.int/news-room/fact-sheets/detail/obesity-and-overweight
5. Also, not a good start with non-grammatical construction: "had overweight or obesity" and "Adolescents with overweight or obesity". Overweight is an adjective not a noun. You can not have overweight. You can be overweight.
Author Response
Reviewer 1:
Overall a nice study and generally well presented manuscript that details a novel approach to developing a text messaging-based system to promote healthy behaviours, especially proper nutrition. Good job.Response: Thank you.
The primary concerns with this manuscript is that it needs another round of thorough editing, and perhaps consideration of adding a bit more analysis related to the distribution of the 107 messages across the 4 content priority areas.
Response: We have proofread and edited the manuscript.
There are numerous typographical and grammatical errors throughout, including in Tables 1 and 3. These need to be cleaned up.
Response: The topographical errors in Table 3 relate to the URLs, Australian spelling, hashtags or puns used in the text messages. We did not identify any grammatical errors in Table 1.
Table 2. Participant characteristics (n=40) The line that reads: "Ethnicity English speaking(a) 21" It seems odd that Ethnicity and English language proficiency are treated as the same. Perhaps the label Ethnicity should be changed. Also, in the text, English language proficiency was identified as an inclusion criterion, but in the table only 21 of the 25 adolescents are identified as speaking English, with a footnote identifying other languages spoken at home. There is a mismatch between each item on this line of the table, and a mismatch with the main text. Needs to be fixed.
Response: We have updated the text and table to correct the mismatch. Please refer to Table 2 and page 3, line 110.
Figure 1 does a nice job of representing the 4 main content priority areas for the text messaging intervention. It would be additionally helpful to see a table or other representation of how the 107 messages in the final selection are distributed across these 4 areas, and how the rejected messages are distributed. Might shed some light on the relative difficulty of messaging in each of these 4 areas.
Response: Thank you for the suggestion. We have included the distribution of excluded messages across the 4 main content priority areas. Please refer to track changes in the manuscript.
The citation at the very beginning seems odd. It does not say where these 42 million children are. There are over 13.7 million overweight or obese children and adolescents under 19 years of age just in the US. The WHO estimates that there were 340 million children and adolescents worldwide in 2016 who were overweight. https://www.cdc.gov/obesity/data/childhood.html https://www.who.int/news-room/fact-sheets/detail/obesity-and-overweight
Response: Thank you for picking up this error. The correct number is 337 million and the reference is correct. We have updated the manuscript. Please refer to track changes in the manuscript.
Also, not a good start with non-grammatical construction: "had overweight or obesity" and "Adolescents with overweight or obesity". Overweight is an adjective not a noun. You can not have overweight. You can be overweight.
Response: We respectfully suggest person‐first language is used throughout our manuscript. Please refer to:
Carl J Palad and Fatima Cody Stanford, Use of people-first language with regard to obesity, The American Journal of Clinical Nutrition, 10.1093/ajcn/nqy076, 108, 1, (201-203), (2018). Armstrong, S. C., Puhl, R., Skinner, A. C., and Kratka, A. (2018) Person‐first language in pediatric obesity research. Pediatric Obesity, 13: 130. doi: 10.1111/ijpo.12211.
Reviewer 2 Report
This study aimed to co-design a bank of text messages that are evidence-based, acceptable, and engaging for adolescents, 13 to 18 years at risk of obesity. I applaud the authors for conducting this important work in a rigorous manner. I only have a few clarifying questions/comments that I hope can strengthen the manuscript when reviewed.
The authors indicate in their objectives that their research targeted adolescents at risk for obesity. It is unclear whether the authors are referring to adolescents with suboptimal lifestyle behaviors only or to a broader definition of “at risk”, such as familial obesity. I encourage the authors to remove this specificity, particularly since they do not report on participant characteristics in detail. A key component to the authors’ research was the involvement of an adolescent research assistant. I would like to request more detail on this adolescent: how were he/she selected? Was this an advertised position? How many interests did the authors receive? What are the adolescents’ demographic and anthropometric characteristics? Did they undergo any research training? I would also appreciate a lens on patient-oriented research, commenting on engagement of the adolescent and focusing on patient-identified priorities in research to improve patient outcomes. One of the response options re: age suitability of the individual text messages limits ages to 13-18-year-olds. I wonder if some of the text messages would have been rated beyond this range if given the option? Parents are mentioned sporadically during the paper as participants, however there is no explicit mention of details surrounding this, which would be essential since the objectives focus on adolescent ratings. In the methods section, the authors state that each text message was reviewed six times by three adolescents and by three professionals. This is slightly confusing to me. Is this different than the the 40 participants, including 15 professionals, who reviewed the text message bank? How were these participants selected/recruited from? More info on this should be provided in the methods section, similarly, more detail on those involved in reviewing the initial text message bank (e.g., removing duplicates – their credentials, etc.). Platform testing – who were the end users of the mobile platforms who provided their replies? Were these participants from the EMPOWER-SMS? I would like to request a table (appendix) including all of the text messages with each targeted corresponding behavioral change technique.Author Response
Reviewer 2:
This study aimed to co-design a bank of text messages that are evidence-based, acceptable, and engaging for adolescents, 13 to 18 years at risk of obesity. I applaud the authors for conducting this important work in a rigorous manner. I only have a few clarifying questions/comments that I hope can strengthen the manuscript when reviewed.Response: Thank you for your comment.
The authors indicate in their objectives that their research targeted adolescents at risk for obesity. It is unclear whether the authors are referring to adolescents with suboptimal lifestyle behaviors only or to a broader definition of “at risk”, such as familial obesity. I encourage the authors to remove this specificity, particularly since they do not report on participant characteristics in detail.
Response: We agree. We have removed the ‘at risk of obesity’ from the aim of the study. Please refer to track changes in the manuscript.
A key component to the authors’ research was the involvement of an adolescent research assistant. I would like to request more detail on this adolescent: how were he/she selected? Was this an advertised position? How many interests did the authors receive? What are the adolescents’ demographic and anthropometric characteristics? Did they undergo any research training? I would also appreciate a lens on patient-oriented research, commenting on engagement of the adolescent and focusing on patient-identified priorities in research to improve patient outcomes.
Response: Thank you for your comment. The adolescent research assistant was a member of the research team and not a participant in the study. They were recruited through our networks and the only demographic recruitment criteria was age (i.e. an adolescent 10-19 years of age). We have included their age in the manuscript with permission. We did not hire based on any other demographic or anthropometric characteristics. The adolescent research assistant completed research training specific to the current study offered by the research team and general research training offered by the university. Please refer to track changes in the manuscript.
One of the response options re: age suitability of the individual text messages limits ages to 13-18-year-olds. I wonder if some of the text messages would have been rated beyond this range if given the option?
Response: This question was optional, and a respondent could none or all ages that apply. We have clarified in the manuscript. Please refer to track changes in the manuscript.
Parents are mentioned sporadically during the paper as participants, however there is no explicit mention of details surrounding this, which would be essential since the objectives focus on adolescent ratings.
Response: Two of the professional reviewers also identified as parents of adolescents 13-18 years of age under the ‘expertise’ question. However, we note this is confusing for the reader and we have updated the manuscript to remove ‘parents’ as this was not an inclusion criterion. Please refer to track changes in the manuscript.
In the methods section, the authors state that each text message was reviewed six times by three adolescents and by three professionals. This is slightly confusing to me. Is this different than the 40 participants, including 15 professionals, who reviewed the text message bank? How were these participants selected/recruited from? More info on this should be provided in the methods section, similarly, more detail on those involved in reviewing the initial text message bank (e.g., removing duplicates – their credentials, etc.).
Response: The 40 participants each reviewed between 15 to 30 individual text messages. We have updated the manuscript including “Each participant reviewed between 15 to 30 individual text messages.” We have also updated the manuscript to include how adolescents were recruited from the wider community “from the wider community via emails to youth groups in local health district and youth specific contacts from the research team’s networks.” Please refer to track changes in the manuscript.
Platform testing – who were the end users of the mobile platforms who provided their replies? Were these participants from the EMPOWER-SMS?
Response: The end users were members of the research team. The 2 programs were tested in conjunction. We have removed the name and reference of the other study. Please refer to track changes in the manuscript.
I would like to request a table (appendix) including all of the text messages with each targeted corresponding behavioral change technique.
Response: The RCT is currently underway and the text message bank will remain confidential until the study is completed.
Reviewer 3 Report
Overall, this is interesting and novel work that adds to the literature. I question the multiple inclusions of work that is not published but is under peer review.
Line 43-44: Add “the ages of” or “aged” after the word between
Line 44 – 47: Switch these 2 sentences to show a progression. Discuss issues with adolescents, young adults and end with a consequence in adulthood.
Line 51: Does research tell us that interventions that have been delivered in-person decrease acceptability?
Line 51-52: Do you mean in all intervention studies? There is a body of evidence in Community-Based Participatory Research that focuses on the inclusion of stakeholders (adolescents) in the research.
Line 54: Consider changing the word “tokenistic”
Line 59: Is this true of adolescents' communication with all groups (e.g., peers vs. adults)?
Line 60: Text messages can incur a cost particularly if a data plan does not include an unlimited text message allowance
Line 65-68: All eight articles should be cited. Also, there are several instances where work has not been published is mentioned. One reference may be acceptable but several references leave questions to the timing of the current manuscript. If this information remains it can be changed to “A review” and remove “under peer review”
Line 70-72: See note referenced above regarding research “under peer review”. If lines 70 through 78 are not needed to describe the current research discussed in the manuscript suggest removing.
Line 95-96: What was the duration of the workshop?
Line 112: Should read adolescent reviewers, add “aged” after (i). This should be consistent throughout the text.
Line 113: Strike “provide” after provided
Line 112-114: Did the participants <18 provide assent? Please clarify.
Line 114: Adolescents reviewers should be changed to Adolescent reviewers. There are additional instances where this should be changed
Line 123: Consider indication (out of 15 points) instead of (/15)
Line 125: The text mentions parent reviewers but parents are not mentioned previously. Please clarify.
Line 127: Should “and professionals” be added after adolescents (>12 by adolescents and professionals)
Line 130: What is mean by led?
Line 134-136: Add a comma after “2”, remove “to” before "develop", and add a comma after "intervention".
Line 146: Consider mentioning the commercial platform used.
Line 160: Is the information about calcium necessary? Nowhere else is calcium mentioned.
Line 160: Consider including a connector such as “Additionally, they are the
Line 162-170: Consider following the same arch as proceeding paragraph. For example, start with adolescents and then discuss adult behaviors
Line 170: Cite information
Line 180: Remove “d” at the end of decrease
Line 185: Cite after diet quality
Line 188: Paragraph opens with factors associated with obesity. The following sentence says the same thing. Could the paragraph open with “Other behavioral factors are associated with overweight/obesity”. In the next paragraph instead of using “associated” again consider a different word.
Line 200: Cite sentence
Line 204: Add “as” - “change such as time”
Line 210-218: I’m unsure if the authors of the current study are implying they used the framework from the systematic review mentioned (citation 51). This could be made clearer. For example, line 213 starts “The reviewer identified six BCTs”. Is this the reviewer of the cited work or the current work. Also, Line 216 indicates “One effective BCT for prevention interventions was identified….” How is this component different from the six BCTs mentioned inline 213. The information could be made clearer.
Line 222: How long was the workshop (one day, 3 hours)?
Line 264: Table 3 – The first 4 messages include the score however, the next 4 do not. Include scores for consistency.
Line 281: Consider listing component with the number after. For example, “Text messages address environment restricting (n=14), stress management or emotional control training (n=14),…”
Line 284: Remove “As well there are”. Capitalize Six and move “sent once per month” to the end of the sentence.
Line 287: Include the type of readability score. “The mean Flesch-Kincaid readability ease score…”
Line 294-300: The introduction/inclusion of the EMPOWER-SMS program seems unnecessary.
Line 317: Again indicates a paper under review. Can the original studies be cited instead?
Line 317: Remove “Also” and capitalize “There”
Line 322-323: The sentence seems similar to the previous one. Also, are the findings to which you refer, findings from the 5 studies or findings from this study?
Line 326: Change “of the findings” to “of these findings”
Line 332-336: Community-based participatory research provides guidance to incorporating all stakeholders including adolescents.
Line 336 – 339: Run-on sentence. Consider reworking sentences.
Author Response
Reviewer 3:
Overall, this is interesting and novel work that adds to the literature. I question the multiple inclusions of work that is not published but is under peer review.Response: Thank you for your detailed feedback. Our protocol paper has now been accepted for publication and we have included the reference. Our systematic review is still in peer review. As suggested in the reviewers comment below, we have referenced the 8 included papers in our systematic review. Please refer to track changes in the manuscript.
Line 43-44: Add “the ages of” or “aged” after the word between
Response: We have included “the ages of” after the word between. Please refer to track changes in the manuscript.
Line 44 – 47: Switch these 2 sentences to show a progression. Discuss issues with adolescents, young adults and end with a consequence in adulthood.
Response: We have switched the 2 sentences. Please refer to track changes in the manuscript.
Line 51: Does research tell us that interventions that have been delivered in-person decrease acceptability?
Response: We have deleted ‘acceptability’ as it was ambiguous. Please refer to track changes in the manuscript.
Line 51-52: Do you mean in all intervention studies? There is a body of evidence in Community-Based Participatory Research that focuses on the inclusion of stakeholders (adolescents) in the research.
Response: We have clarified “In obesity prevention and management intervention studies” track changes in the manuscript.
Line 54: Consider changing the word “tokenistic”
Response: Tokenistic is the correct word.
Line 59: Is this true of adolescents' communication with all groups (e.g., peers vs. adults)?
Response: We have updated to this statement. “Amongst all smartphone capabilities, text messages remain a preferred form of communication for adolescents to communicate with their peers.” Please refer to track changes in the manuscript.
Line 60: Text messages can incur a cost particularly if a data plan does not include an unlimited text message allowance
Response: This statement is referring to receiving text messages. Receiving a text message does not incur a cost.
Line 65-68: All eight articles should be cited. Also, there are several instances where work has not been published is mentioned. One reference may be acceptable but several references leave questions to the timing of the current manuscript. If this information remains it can be changed to “A review” and remove “under peer review”
Response: Our systematic review and protocol paper were submitted for publication in August and September, respectively. Our protocol paper is now accepted for publication and our systematic review is still under peer review. We have updated the manuscript with the reference for the protocol and cited the eight articles and changed to “A review” as suggested.
Line 70-72: See note referenced above regarding research “under peer review”. If lines 70 through 78 are not needed to describe the current research discussed in the manuscript suggest removing.
Response: Please see previous responses.
Line 95-96: What was the duration of the workshop?
Response: The workshop was 2-hours, we have updated the manuscript. Please refer to track changes in the manuscript.
Line 112: Should read adolescent reviewers, add “aged” after (i). This should be consistent throughout the text.
Response: We have updated the manuscript. Please refer to track changes throughout the manuscript.
Line 113: Strike “provide” after provided
Response: Thank you, we have removed the duplication. Please refer to track changes in the manuscript.
Line 112-114: Did the participants <18 provide assent? Please clarify.
Response: Yes. We have clarified in the text to state that if <18 years of age, they provided written informed consent and additional written informed consent from their parents or guardians. Please refer to track changes in the manuscript.
Line 114: Adolescents reviewers should be changed to Adolescent reviewers. There are additional instances where this should be changed
Response: Thank you, we have fixed the 2 errors. Please refer to track changes in the manuscript.
Line 123: Consider indication (out of 15 points) instead of (/15)
Response: We have updated to out of 15 points. Please refer to track changes in the manuscript.
Line 125: The text mentions parent reviewers but parents are not mentioned previously. Please clarify.
Response: Two of the professional reviewers also identified as parents of adolescents 13-18 years of age under the ‘expertise’ question. However, we note this is confusing for the reader and we have updated the manuscript to remove ‘parents’as this was not an inclusion criterion. Please refer to track changes throughout the manuscript.
Line 127: Should “and professionals” be added after adolescents (>12 by adolescents and professionals)
Response: Yes. We have updated the manuscript. Please refer to track changes in the manuscript.
Line 130: What is mean by led?
Response: We have removed the word ‘led’ and updated the sentence. Please refer to track changes in the manuscript.
Line 134-136: Add a comma after “2”, remove “to” before "develop", and add a comma after "intervention".
Response: We have updated this sentence. Please refer to track changes in the manuscript.
Line 146: Consider mentioning the commercial platform used.
Response: We have included the name of the commercial platform. Please refer to track changes in the manuscript.
Line 160: Is the information about calcium necessary? Nowhere else is calcium mentioned.
Response: We agree and have deleted this information. Please refer to track changes in the manuscript.
Line 160: Consider including a connector such as “Additionally, they are the
Response: We have included a connector. Please refer to track changes in the manuscript.
Line 162-170: Consider following the same arch as proceeding paragraph. For example, start with adolescents and then discuss adult behaviors
Response: We have updated this paragraph. Please refer to track changes in the manuscript.
Line 170: Cite information
Response: We have included a reference for “However, diets high in fruits and vegetables are essential for overall health and are independently associated with a decreased risk of other non-communicable diseases such as cardiovascular disease, stroke, and some cancers [40]”
Wang, X.; Ouyang, Y.; Liu, J.; Zhu, M.; Zhao, G.; Bao, W.; Hu, F.B. Fruit and vegetable consumption and mortality from all causes, cardiovascular disease, and cancer: Systematic review and dose-response meta-analysis of prospective cohort studies. BMJ (Clinical research ed.) 2014, 349, g4490.
Line 180: Remove “d” at the end of decrease
Response: We have removed “d” at the end of decrease. Please refer to track changes in the manuscript.
Line 185: Cite after diet quality
Response: We have removed the semi-colon from the sentence and provided a reference. Please refer to track changes in the manuscript.
Line 188: Paragraph opens with factors associated with obesity. The following sentence says the same thing. Could the paragraph open with “Other behavioral factors are associated with overweight/obesity”. In the next paragraph instead of using “associated” again consider a different word.
Response: We have updated the opening sentence for this paragraph. Please refer to track changes in the manuscript.
Line 200: Cite sentence
Response: We have included the 3 citations at the end of this sentence. Please refer to track changes in the manuscript.
Line 204: Add “as” - “change such as time”
Response: We have added “as” - “change such as time” Please refer to track changes in the manuscript.
Line 210-218: I’m unsure if the authors of the current study are implying they used the framework from the systematic review mentioned (citation 51). This could be made clearer. For example, line 213 starts “The reviewer identified six BCTs”. Is this the reviewer of the cited work or the current work. Also, Line 216 indicates “One effective BCT for prevention interventions was identified….” How is this component different from the six BCTs mentioned inline 213. The information could be made clearer.
Response: We have clarified this statement. Please refer to track changes in the manuscript.
Line 222: How long was the workshop (one day, 3 hours)?
Response: The workshop was 2-hours. We have clarified in the manuscript. Please refer to track changes in the manuscript.
Line 264: Table 3 – The first 4 messages include the score however, the next 4 do not. Include scores for consistency.
Response: We have included the scores for all the text messages in Table 3. Please refer to
Line 281: Consider listing component with the number after. For example, “Text messages address environment restricting (n=14), stress management or emotional control training (n=14),…”
Response: We have listed the component with the number after. Please refer to track changes in the manuscript.
Line 284: Remove “As well there are”. Capitalize Six and move “sent once per month” to the end of the sentence.
Response: We have updated this sentence. Please refer to track changes in the manuscript.
Line 287: Include the type of readability score. “The mean Flesch-Kincaid readability ease score…”
Response: We have updated this sentence. Please refer to track changes in the manuscript.
Line 294-300: The introduction/inclusion of the EMPOWER-SMS program seems unnecessary.
Response: The 2 programs were tested in conjunction. We have removed the name and reference of the other study. Please refer to track changes in the manuscript.
Line 317: Again indicates a paper under review. Can the original studies be cited instead?
Response: We have included the eight studies as references and updated the manuscript. Please refer to previous responses.
Line 317: Remove “Also” and capitalize “There”’
Response: We have removed Also” and capitalize “There” Please refer to track changes in the manuscript.
Line 322-323: The sentence seems similar to the previous one. Also, are the findings to which you refer, findings from the 5 studies or findings from this study?
Response: We have clarified this sentence. Please refer to track changes in the manuscript.
Line 326: Change “of the findings” to “of these findings”
Response: We have changed “of the findings” to “of these findings” Please refer to track changes in the manuscript.
Line 332-336: Community-based participatory research provides guidance to incorporating all stakeholders including adolescents.
Response: The systematic review cited, suggested different frameworks are required for different stakeholder groups.
Line 336 – 339: Run-on sentence. Consider reworking sentences.
Response: We have reworked this sentence. Please refer to track changes in the manuscript.
Reviewer 4 Report
This is a very interesting and highly relevant manuscript. Adolescence as an opportunity to intervene to prevent long term health conditions, is gaining greater prominence. Yet, research into development of interventions with adolescents is relatively sparse. The importance of co-creating interventions with target audiences is well known, and yet, there are only a few research publications detailing this work with this specific target audience. So I applaud this piece of work and look forward to reading the other manuscripts referenced as ‘under peer review’. I have a few minor comments detailed below, but also a couple of more major comments:
On page 3, in the methods section you describe your adolescent panel as being recruited from “an outpatient weight management clinic in a public hospital or from the wider community”. In table 2 ‘participant demographics’, I would like to see the numbers of adolescent participants from these 2 groups. I would also like more information about what the ‘wider community’ is and where/how you recruited these participants. I would predict that there may be a difference in the perspectives of the text messages from adolescents who are attending a weight management clinic and those from the wider community. Table 3 – you give details about rating level for messages which score highly, but not consistently for those which were subsequently excluded – maybe you could include an additional column with the actual rating for each statement you discuss, I would also like to see the breakdown of that rating between the three participant groups - you talk about adolescents and parents as one group and professionals as another, and sometimes just adolescents on their own – it feels a bit muddled and difficult to understand and I would like to see the perspectives of all 3 groups separately (and also to see if the adolescents from the weight management clinic had a different perspective to the adolescents from the wider community) – I think this is essential to really understand how this work is applicable more broadly for other researchers. I was interested in your justification for the focus on specific BCTs – I feel you have made some major decisions about which BCTs to focus on, on the basis of 1 paper, which itself comments on its limitations and cautions that other BCTs shouldn’t be summarily dismissed. Pragmatic choices have to be made around where to focus attention, but I think this warrants a discussion of the limitations this could introduce to this work, but also for the RCT. There was only a very brief exploration of the limitations of this work in the discussion, which I feel is a weakness of this manuscript. You have small numbers of adolescents participating, you highlight that these were recruited using a convenience sampling method, but without further details about how this was carried out, it’s hard to judge how representative of a general population these might be – were they recruited through schools? through youth groups, through sports groups? Additionally, a subset of these adolescents were recruited from an outpatient weight management clinic and so potentially have a particular perspective around the messaging. You have a high female proportion of participants, again, as you note, this could bias the insights you are getting – did you do some thematic work on the participant responses? Were there difference in the responses from male adolescents compared to female? More detail here would be useful.Minor comments:
Introduction
Line 42 – Are these global figures?
Line 65-67 – ‘Our recent systematic review found only eight interventions that utilized text messages for obesity prevention and management in adolescents which utilized text messages (under peer review).’ –you have duplicated the text ‘which utilized text messages’.
Line 80 – What is the justification for choosing 13-18 yrs as your adolescent range, why not 10-19yrs (WHO) or 10-24yrs (maybe this is explained in another manuscript?).
Results
Line 180 – should be ‘randomized controlled trial’
Line 180 – ‘decreased’ should be ‘decrease’
Line 184 – ‘Emerging research suggests to improve an individual’s…..’
Line 204 – ‘addressing barriers to behavior change such as time management’
Line 227 – ‘at random times to reduce pattern detection’
Table 2 – How many adolescents were from the wider community and how many from the weight management outpatient clinic. How many parents – these aren’t included at all in this table and should be another separate section
Line 254 – ‘The reasons for inclusion’.
Line 279 – ‘prompting the adolescent to repeat the target behaviors’
Line 284 – ‘As well, there are…’
Line 288 – ‘indicating the text message bank, of on average…’
Line 288 – What age range is seventh-grade (thinking about your international audience)
Discussion
Line 305 – you say ‘this study worked with adolescents from the wider community, as well as an adolescent research assistant’ – what about the adolescents from the weight management outpatient clinic?
Line 316 – ‘…using text messages for programme delivery, report outcomes….’
Line 316 – ‘…the process of text messages development…’
Author Response
Reviewer 4:
This is a very interesting and highly relevant manuscript. Adolescence as an opportunity to intervene to prevent long term health conditions, is gaining greater prominence. Yet, research into development of interventions with adolescents is relatively sparse. The importance of co-creating interventions with target audiences is well known, and yet, there are only a few research publications detailing this work with this specific target audience. So I applaud this piece of work and look forward to reading the other manuscripts referenced as ‘under peer review’. I have a few minor comments detailed below, but also a couple of more major comments.Response: Thank you for your comment.
On page 3, in the methods section you describe your adolescent panel as being recruited from “an outpatient weight management clinic in a public hospital or from the wider community”. In table 2 ‘participant demographics’, I would like to see the numbers of adolescent participants from these 2 groups. I would also like more information about what the ‘wider community’ is and where/how you recruited these participants. I would predict that there may be a difference in the perspectives of the text messages from adolescents who are attending a weight management clinic and those from the wider community.
Response: Thank you for your comment. We did not include a question in the demographic survey to distinguish where an adolescent participant was recruited from and we did not ask participants for any anthropometric characteristics. We have included this as a limitation. Please refer to track changes in the manuscript.
Table 3 – you give details about rating level for messages which score highly, but not consistently for those which were subsequently excluded – maybe you could include an additional column with the actual rating for each statement you discuss, I would also like to see the breakdown of that rating between the three participant groups - you talk about adolescents and parents as one group and professionals as another, and sometimes just adolescents on their own – it feels a bit muddled and difficult to understand and I would like to see the perspectives of all 3 groups separately (and also to see if the adolescents from the weight management clinic had a different perspective to the adolescents from the wider community) – I think this is essential to really understand how this work is applicable more broadly for other researchers.
Response: We have included the scores for each of the text messages in Table 3. We have included in the limitations section that the process evaluation in the RCT will provide wider views on the acceptability of the text message program in adolescents with overweight. Please refer to track changes in the manuscript.
I was interested in your justification for the focus on specific BCTs – I feel you have made some major decisions about which BCTs to focus on, on the basis of 1 paper, which itself comments on its limitations and cautions that other BCTs shouldn’t be summarily dismissed. Pragmatic choices have to be made around where to focus attention, but I think this warrants a discussion of the limitations this could introduce to this work, but also for the RCT. There was only a very brief exploration of the limitations of this work in the discussion, which I feel is a weakness of this manuscript. You have small numbers of adolescents participating, you highlight that these were recruited using a convenience sampling method, but without further details about how this was carried out, it’s hard to judge how representative of a general population these might be – were they recruited through schools? through youth groups, through sports groups? Additionally, a subset of these adolescents were recruited from an outpatient weight management clinic and so potentially have a particular perspective around the messaging. You have a high female proportion of participants, again, as you note, this could bias the insights you are getting – did you do some thematic work on the participant responses? Were there difference in the responses from male adolescents compared to female? More detail here would be useful.
Response: We agree this is a limitation and have included this in the limitations section. Obesity prevention in adolescents is an under researched area and we have used the best available evidence. The seven BCTs selected for the text messages are based on one systematic review, which included the intervention content of 17 individual intervention studies assessing BMI outcomes at six months. As mentioned in the manuscript, additional support for behaviour change will be provided through communication with the health counsellor using complementary theoretical approaches including motivational interviewing, goal setting, self-monitoring and barrier identification and problem solving. Please refer to track changes in the manuscript.
Line 42 – Are these global figures?
Response: Thank you for picking up this error. The correct number is 337 million and the reference is correct. Please refer to track changes in the manuscript.
Line 65-67 – ‘Our recent systematic review found only eight interventions that utilized text messages for obesity prevention and management in adolescents which utilized text messages (under peer review).’ –you have duplicated the text ‘which utilized text messages’.
Response: Thank you for picking up this error. We have deleted the duplication. Please refer to track changes in the manuscript.
Line 80 – What is the justification for choosing 13-18 yrs as your adolescent range, why not 10-19yrs (WHO) or 10-24yrs (maybe this is explained in another manuscript?).
Response: The protocol manuscript has now been accepted for publication. The justification for 13-18 years of age is justified in this manuscript. We have now referenced this publication.
Line 180 – should be ‘randomized controlled trial’
Response: Thank you for picking up this error. We have fixed. Please refer to track changes in the manuscript.
Line 180 – ‘decreased’ should be ‘decrease’
Response: We have corrected this error. Please refer to track changes in the manuscript.
Line 184 – ‘Emerging research suggests to improve an individual’s…..’
Response: We have corrected this error. Please refer to track changes in the manuscript.
Line 204 – ‘addressing barriers to behavior change such as time management’
Response: We have corrected this error. Please refer to track changes in the manuscript.
Line 227 – ‘at random times to reduce pattern detection’
Response: We have corrected this error. Please refer to track changes in the manuscript.
Table 2 – How many adolescents were from the wider community and how many from the weight management outpatient clinic. How many parents – these aren’t included at all in this table and should be another separate section
Response: Thank you for your comment. We did not include a question in the demographic survey to distinguish where an adolescent participant was recruited from and we did not ask participants for any anthropometric characteristics. We have included this as a limitation. Please refer to track changes in the manuscript.
Line 254 – ‘The reasons for inclusion’.
Response: We have corrected this error. Please refer to track changes in the manuscript.
Line 279 – ‘prompting the adolescent to repeat the target behaviors’
Response: We have corrected this error. Please refer to track changes in the manuscript.
Line 284 – ‘As well, there are…’
Response: We have corrected this error. Please refer to track changes in the manuscript.
Line 288 – ‘indicating the text message bank, of on average…’
Response: We have added a comma. Please refer to track changes in the manuscript.
Line 288 – What age range is seventh-grade (thinking about your international audience)
Response: Thank you. We have included the corresponding ages for each grade mentioned in the manuscript. Please refer to track changes in the manuscript.
Line 305 – you say ‘this study worked with adolescents from the wider community, as well as an adolescent research assistant’ – what about the adolescents from the weight management outpatient clinic?
Response: We have included adolescents from the weight management clinic in this sentence. Please refer to track changes in the manuscript.
Line 316 – ‘…using text messages for programme delivery, report outcomes….’
Response: We have corrected this error. Please refer to track changes in the manuscript.
Line 316 – ‘…the process of text messages development…’
Response: We have corrected this error. Please refer to track changes in the manuscript.